# A Model of Interacting Navier–Stokes Singularities

**DOI:** 10.3390/e24070897

**Published:** 2022-06-29

**Authors:** Hugues Faller, Lucas Fery, Damien Geneste, Bérengère Dubrulle

**Affiliations:** 1Service de Physique de l’État Condensé, CNRS UMR 3680, CEA, Université Paris-Saclay, 91190 Gif-sur-Yvette, France; hugues.faller__recherche@normalesup.org (H.F.); damien.geneste@cea.fr (D.G.); berengere.dubrulle@cea.fr (B.D.); 2Department of Physics, Ecole Normale Supérieure de Lyon, 69364 Lyon, France

**Keywords:** turbulence, singularity, non-equilibrium dynamics

## Abstract

We introduce a model of interacting singularities of Navier–Stokes equations, named pinçons. They follow non-equilibrium dynamics, obtained by the condition that the velocity field around these singularities obeys locally Navier–Stokes equations. This model can be seen as a generalization of the vorton model of Novikov that was derived for the Euler equations. When immersed in a regular field, the pinçons are further transported and sheared by the regular field, while applying a stress onto the regular field that becomes dominant at a scale that is smaller than the Kolmogorov length. We apply this model to compute the motion of a pair of pinçons. A pinçon dipole is intrinsically repelling and the pinçons generically run away from each other in the early stage of their interaction. At a late time, the dissipation takes over, and the dipole dies over a viscous time scale. In the presence of a stochastic forcing, the dipole tends to orientate itself so that its components are perpendicular to their separation, and it can then follow during a transient time a near out-of-equilibrium state, with forcing balancing dissipation. In the general case where the pinçons have arbitrary intensity and orientation, we observe three generic dynamics in the early stage: one collapse with infinite dissipation, and two expansion modes, the dipolar anti-aligned runaway and an anisotropic aligned runaway. The collapse of a pair of pinçons follows several characteristics of the reconnection between two vortex rings, including the scaling of the distance between the two components, following Leray scaling $\sqrt{t_{c}-t}$.

## 1. Introduction

Snapshots of dissipation or enstrophy in turbulent fluids show us that small scales are intermittent, localized and irregular. Mathematical theorems constrain the degree of irregularity of such structures that are genuine singularities of the incompressible Navier–Stokes provided their spatial $L^{3}$-norm is unbounded (for a review of various regularity criteria, see [1]). On the other hand, dissipation laws of turbulent flows suggest that they may be at most Hölder continuous with h<1/3 [2] and of diverging vorticity in the inviscid limit. This observation has motivated several theoretical construction of turbulent Navier–Stokes small scale structures or weak solutions of Euler equations, using singular or quasi-singular entities based e.g., on atomic like structures [3], Beltrami flows [4], Mikado flows [5], spirals [6,7], vortex filaments [8], or Lagrangian particles [9].

These constructions have fueled a long-standing analytical framework of turbulence, allowing the modeling of proliferating and numerically greedy small scales by a countable (and hopefully numerically reasonable) number of degrees of freedom, provided by characteristics of the basic entities.

A good example of the possibilities offered by such a singular decomposition is provided by the 3D vorton description of Novikov [10]. In this model, the vorticity field is decomposed into *N* discretized singularities infinitely localized (via a δ function) at points $r_{\alpha }$, (α=1,⋯,N), each characterized by a vector $\gamma_{\alpha }$ providing the intensity and the axis of rotation of motions around such singularities. The singularities are not fixed but move under the action of the velocity field and velocity strain induced by the other singularities, so as to respect conservation of circulation. Around the singularity, the velocity field is not of divergence free, so that the vortons are akin to hydrodynamical monopoles interacting at long-range through a potential decaying like $1/r^{2}$. The model was adapted to enable numerical simulations of interacting vorticity rings or filaments by considering a divergence-free generalization of the vortons [11]. Quite remarkably, the Vorton model results in vortex reconnection, even though no viscosity is introduced in the numerical scheme [12]. Whether the effective viscosity is due to intense vortex stretching [13], or to properties of vortex alignment during reconnection [12], is still debated.

From a mathematical point of view, the vorton model cannot be considered as a fully satisfying description of singularities of Navier–Stokes for two reasons. First, the vortons do not constitute exact weak solutions of the 3D Euler or Navier–Stokes equations [14,15,16], which somehow makes them less attractive than point vortices that are weak solutions of 2D Euler equations [14]. Second, vortons do not respect the scaling invariance of Navier–Stokes, which imposes that the velocity field should scale like 1/r. Indeed, through the Biot–Savart law, we see that a Dirac vortex field induces a velocity scaling like $1/r^{2}$, where *r* is the distance to singularity.

Motivated by this observation, we introduce in this paper a modification of the vorton model that is built upon weak solutions of Navier–Stokes equations, and which respects scale invariance of the Navier–Stokes equations, and allows simple dynamical description of the evolution of the basic entities, hereafter named pinçons.

After useful generalities (Section 2.1), we introduce the pinçon model (Section 2.2) and describe their properties in Section 2.3. We introduce the non-equilibrium dynamics of pinçons in Section 2.4 and Section 2.5. We then solve the equations in Section 3, starting with the special case of a dipole in Section 3.2 and Section 3.3, and concluding with the general case in Section 3.4. A discussion follows in Section 4.

## 2. Pinçon Model

### 2.1. Generalities and Ideas behind the Pinçon Model

Consider a velocity field U obeying the Navier–Stokes equations. Then, it is well known that the coarse-grained field ${\bar{U}}^{\ell }$ obeys the Navier–Stokes equations forced by the “turbulent force” due to the Reynolds stress $\nabla \cdot \left({\bar{U}}^{\ell }{\bar{U}}^{\ell }-{\bar{UU}}^{\ell }\right)$. Numerical and experimental observations also show that, as ℓ→0, this turbulent force becomes more and more intermittent, made of isolated patches of finite values, in a sea of zero values. The size of the isolated patches shrinks with decaying *ℓ*. In our experiment, we have observed that such patches do persist even when *ℓ* is of the order of the Kolmogorov scale $\eta_{K}$, and have correlated such patches with the existence of nonzero local energy transfers at such location.

This means that numerical simulations of Navier–Stokes must have a resolution much smaller than $\eta_{K}$ in order to fully resolve not only velocity gradients [17] but also local energy transfers and dissipation [18]. The numerical price to pay is high, especially at large Reynolds number, and a lot of computing time is wasted in the tracking of increasingly thinner regions of space.

To avoid this, a natural idea is to split the fluid in two component: one, continuous, representing the coarse-grained fluid ${\bar{U}}^{\ell }$, for a scale $\ell =\ell_{c}$ to be determined later, and one, discrete, representing the isolated patches of unresolved fluid that are fed by the turbulent force, then carry and dissipate the corresponding energy with dynamics to be determined later. In this view, the small scales must therefore be represented by modes that are representative of the small scale behaviour of Navier–Stokes. Given that the aim is to be able to describe the whole range of scales $\ell <\ell_{c}$, it is natural to consider self-similar solutions of Navier–Stokes, i.e., solutions that are invariant by the (Leray) rescaling $U\left(x,\;t\right)\to \lambda^{-1}U\left(x/\lambda ,\;t/\lambda^{2}\right)$ for any λ [19]. Moreover, to be able to describe the small scales by modes dynamics, we consider self-similar solutions that do not explicitly depend on time, so that all the dynamics will be contained in the time variation of the modes’ parameters. Corresponding solutions then obey
(1)$$\forall \lambda \neq 0,U\left(x\right)=\lambda^{-1}U\left(x/\lambda \right),$$
corresponding to homogeneous Navier–Stokes solutions of degree −1.

As shown by Sverak [20], Theorem 1, the only non-trivial solutions that are smooth in R\{0} are axisymmetric, and correspond to Landau solutions, described in Section 2.2, Equation (5). These solutions obey the stationary Navier–Stokes equations everywhere except at the origin. Specifically, we have in some distributional sense: (2)$$\begin{array}{ccc} \nabla \cdot U & = & 0, \end{array}$$(3)$$\begin{array}{ccc} (U\cdot \nabla )U+\frac{\nabla p}{\rho }-\nu \Delta U & = & \nu^{2}\delta \left(x\right)F, \end{array}$$
where F is a vector of magnitude *F* and given orientation e, providing the axis of symmetry.

Thus, there is not much choice in the building of small scales modes. Here is how they are built, using Landau solutions.

### 2.2. Definition of Pinçon

We introduce the pinçons as individual entities labeled by α, characterized by their position $x_{\alpha }\left(t\right)$, and a non-dimensional vector $\gamma_{\alpha }\left(t\right)$, with $\gamma_{\alpha }=∥\gamma_{\alpha }∥<1$ that produce locally an axisymmetric velocity field around their axis of direction $\gamma_{\alpha }$ given by $\left(p_{\alpha },v_{\alpha }\right)\left(x\right)\equiv \left(p\left(x-x_{\alpha },\gamma_{\alpha }\right),U\left(x-x_{\alpha },\gamma_{\alpha }\right)\right)$ with U and *p* given by: (4)$$\begin{array}{ccc} U(x,\gamma ) & = & 2\nu \left(\frac{1}{\phi }\left(\gamma -\frac{x}{∥x∥}\right)+\left(1-\gamma^{2}\right)\frac{x}{\phi^{2}}\right), \end{array}$$(5)$$\begin{array}{ccc} p(x,\gamma ) & = & -4\rho \nu^{2}\left(\frac{1}{∥x∥\phi }+\frac{1-\gamma^{2}}{\phi^{2}}\right). \end{array}$$
where $p_{\alpha }$ is the associated pressure, ϕ(x,γ)=∥x∥−γ·x and we define $\phi_{\alpha }=\phi \left(x-x_{\alpha },\gamma_{\alpha }\right)$. A few useful properties of ϕ are put in Appendix A. In particular, the velocity field given by Equation (Equation 4) is homogeneous of degree −1 around $x_{\alpha }$, and axisymmetric around the direction of $\gamma_{\alpha }$. Plots of velocity and vorticity around a pinçon are displayed in Figure 1. Close to the singularity, there is a neck pinch of the velocity streamlines, hence their name pinçon. As first shown by Landau [21] (see also [20,22,23,24,25]), the velocity fields $v_{\alpha }$ are solutions of Equations (Equation 2) and (Equation 3) with
(6)$$\begin{array}{ccc} F & = & F\left(\gamma_{\alpha }\right)\frac{\gamma_{\alpha }}{\gamma_{\alpha }},\\ F(\gamma ) & = & 4\pi \left[\frac{4}{\gamma }-\frac{2}{\gamma^{2}}\ln\left(\frac{1+\gamma }{1-\gamma }\right)+\frac{16}{3}\frac{\gamma }{1-\gamma^{2}}\right],\\ \gamma_{\alpha } & = & ∥\gamma_{\alpha }∥. \end{array}$$

We refer the reader to [25] for a rigorous derivation of such result. The function F(γ) is shown in Figure 2a. It starts from 0 at γ=0, corresponding to a state of rest (U=p=0), with a linear behaviour F(γ)=16πγ near the origin, and diverges at γ=1. In the latter case, the velocity field is diverging on a whole semi-axis defined by γ·x=∥x∥, such that ϕ=0. Hence, γ characterizes the intensity of the velocity field of a pinçon.

**Figure 1 entropy-24-00897-f001:**
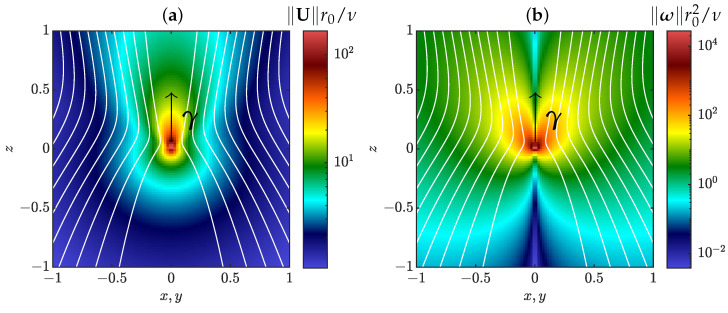
Streamlines (white curves) of velocity field around a pinçon of intensity γ=0.6 in a plane that contains the axis of the pinçon $\gamma =\gamma \;e_{z}$, which is represented by the black arrow. The field is axisymmetric around that axis. The color represents the norm of rescaled velocity (**a**) and vorticity (**b**) fields, where ν is the kinematic viscosity and $r_{0}$ some insignificant length scale. The coordinates x,y,z are also nondimensionalized by $r_{0}$. There is no azimuthal component of velocity, while the vorticity is purely azimuthal with respect to the axis γ.

### 2.3. Properties of Pinçons

#### 2.3.1. Scaling under Coarse-Graining

The velocity field and all its derivatives diverge at the location of the pinçon so they are undefined at such point. Its behavior may, however, be studied near the origin by introducing a suitable test function ψ that is spherically symmetric around x=0, positive of unit integral, $C^{\infty }$ and that decays fast at infinity, considering the regularizations
$${\bar{v_{\alpha }}}^{\ell }\left(x\right)=\int \psi \left(\frac{x-y}{\ell }\right)v_{\alpha }\left(y\right)\frac{dy}{\ell^{3}},$$
where *ℓ* is a small parameter. In the limit ℓ→0, the function $\psi \left(\frac{x}{\ell }\right)$ is peaked around the origin so that, as long as x is far from $x_{\alpha }$, we can estimate: ${\bar{v_{\alpha }}}^{\ell }\left(x\right)\approx v_{\alpha }\left(x\right)$. Consider now the situation where $x=x_{\alpha }$. We have then: $${\bar{v_{\alpha }}}^{\ell }\left(x_{\alpha }\right)=\int \psi \left(\frac{x_{\alpha }-y}{\ell }\right)U\left(y-x_{\alpha },\gamma_{\alpha }\right)\frac{dy}{\ell^{3}}.$$
Applying finally the change of variable $y-x_{\alpha }=\ell z$, using homogeneity properties of U and spherical symmetry of ψ, we have: (7)$$\begin{array}{ccc} {\bar{v_{\alpha }}}^{\ell }\left(x_{\alpha }\right) & = & \frac{1}{\ell }\int \psi \left(z\right)U\left(z,\gamma_{\alpha }\right)dz,\\ & = & \frac{C_{\psi }}{\ell }{\langle v_{\alpha }\rangle }_{B_{1}}\underset{\ell \to 0}{=}O\left(\ell^{-1}\right), \end{array}$$
where $C_{\psi }=4\pi \int r\psi \left(r\right)dr$ and ${\langle v_{\alpha }\rangle }_{B_{1}}$ is the average over the sphere of radius unity. Via Euler theorem, $\nabla v_{\alpha }$ is homogeneous of order −2. By the same reasoning, we then find that ${\bar{\nabla v_{\alpha }}}^{\ell }\left(x_{\alpha }\right)\underset{\ell \to 0}{=}\;O\left(\ell^{-2}\right)$. The same reasoning cannot be applied to $\nabla^{2}v_{\alpha }$ because the integral ∫ψ(r)dr/r does not necessarily converge at the origin. However, the following property holds: ${\bar{\left(v_{\alpha }\cdot \nabla \right)v_{\alpha }}}^{\ell }+\frac{\nabla {\bar{p_{\alpha }}}^{\ell }}{\rho }-\nu \Delta {\bar{v_{\alpha }}}^{\ell }=\frac{\nu^{2}}{\ell^{3}}\psi \left(\frac{x-x_{\alpha }}{\ell }\right)F$, which is $O(\ell^{-3})$.

#### 2.3.2. Potential Vector, Vorticity and Helicity

Using vector calculus identities, one can check that the velocity field around a pinçon derives from the vector potential:(8)$$\begin{array}{ccc} A_{\alpha }\left(x\right) & = & 2\nu \left(x-x_{\alpha }\right)\times \nabla \ln\left(\phi_{\alpha }\right),\\ & = & 2\nu \frac{\gamma_{\alpha }\times \left(x-x_{\alpha }\right)}{\phi_{\alpha }}, \end{array}$$

The vorticity field produced locally around a pinçon can be formally defined by taking the curl of $v_{\alpha }$. The vorticity is parallel to the potential vector and reads:(9)$$\omega_{\alpha }\left(x\right)=4\nu \left(1-\gamma^{2}\right)\frac{\gamma_{\alpha }\times \left(x-x_{\alpha }\right)}{\phi_{\alpha }^{3}},$$

The vorticity field is thus purely azimuthal with respect to the pinçon axis γ, and by axisymmetry, the vorticity lines form rings around the axis. We also notice that the velocity field produced by a pinçon is of zero helicity.

#### 2.3.3. Generalized Momentum and Coarse-Grained Vorticity

We define a generalized momentum $\Pi_{\alpha }$ for the pinçon as an average of the velocity field over a sphere of unit radius (see Appendix B for its computation):(10)$$\begin{array}{ccc} \Pi_{\alpha }\equiv {\langle v_{\alpha }\rangle }_{B_{1}} & = & \nu G\left(\gamma_{\alpha }\right)\frac{\gamma_{\alpha }}{\gamma_{\alpha }},\\ G(\gamma ) & = & \frac{2}{\gamma^{2}}\left[2\gamma -\left(1-\gamma^{2}\right)\ln\left(\frac{1+\gamma }{1-\gamma }\right)\right]. \end{array}$$
By definition, $\Pi_{\alpha }$ provides an estimate of the coarse-grained velocity field at the pinçon position, via ${\bar{v_{\alpha }}}^{\ell }\left(x_{\alpha }\right)=\;C_{\psi }\Pi_{\alpha }/\ell$. Note that $\Pi_{\alpha }$ points in the direction of $\gamma_{\alpha }$. For 0≤γ<1, G(γ) varies smoothly from 0 to 4, starting from a linear behavior G(γ)=8γ/3 at the origin and ending with a vertical tangent at γ=1 (see Figure 2b). Therefore, the function G(γ) is bijective, and there is a one-to-one correspondence between *G* and γ and Π and γ.

Due to the axisymmetry, we have that ${\bar{\omega }}^{\ell }\left(x_{\alpha }\right)=0$, so that the coarse-grained pinçon is vorticity free near its location.

### 2.4. Interaction of a Pinçon with a Regular Field

There are several reasons why the notion of a “single” pinçon does not make a physical sense:(i)the pinçon is dissipative, and requires a force to maintain it; a surrounding fluid can provide the necessary forcing;(ii)the pinçon lives in an infinite universe and does not fulfill the boundary conditions of a realistic system. To be able to use a pinçon in confined systems, we need to add an external velocity field that will take care of the boundary conditions;(iii)if we accept that the pinçon describes the very intermittent part of the energy transfer that cannot be resolved, we must also accept the possibility of coexistence and interaction of several pinçons. If we assume that the pinçons are distinct and that they are regular everywhere except at their position, this amounts again to considering the interaction of a pinçon with a regular field.

Given the nonlinear nature of the Navier–Stokes equations, we cannot superpose two solutions of Navier–Stokes equations (a pinçon and a regular field) to get a solution of Navier–Stokes equations. Instead, we will now assume that the pinçon has a dynamics and find such dynamics by imposing that the superposition of the pinçon and of the regular field locally obeys the Navier–Stokes equations. Specifically, we consider a solution of the shape $v_{\alpha }=U\left(x-x_{\alpha }\left(t\right),\gamma_{\alpha }\left(t\right)\right)$ and F=F(γ(t))e(t), where *F* is a prescribed function, and $x_{\alpha }$, γ=γe, two vectors that parametrize the field $v_{\alpha }$ as a function of *t*. We then introduce $v=v_{R}+v_{\alpha }$, where $v_{R}$ is a velocity field that is regular at the origin, and we impose that v is a solution of Navier–Stokes locally around the singularity at $x_{\alpha }$, i.e., that v is a solution of
(11)$$\partial_{t}{\bar{v}}^{\ell }\left(x_{\alpha }\right)+{\bar{(v\cdot \nabla )v}}^{\ell }\left(x_{\alpha }\right)+\frac{{\bar{\nabla p}}^{\ell }\left(x_{\alpha }\right)}{\rho }-\nu {\bar{\Delta v}}^{\ell }\left(x_{\alpha }\right)=0.$$

Decomposing the velocity field into its regular and irregular part, we see that Equation (Equation 11) generates terms of various orders in *ℓ* that scale according to Table 1. Note that, since $v_{R}$ is a regular field, its coarse-grained version scales like O(1), as well as its derivatives.

Furthermore, we introduce the quantity $\tau^{\ell }={\bar{v_{R}}}^{\ell }{\bar{v_{R}}}^{\ell }-{\bar{v_{R}v_{R}}}^{\ell }$, which is the Reynolds stress contribution due to filtering. This term has a different scaling. Indeed, $\nabla \cdot \tau^{\ell }=\;O\left(\delta v_{\ell }^{2}/\ell \right)$ [26], where $\delta v_{\ell }=\;v_{R}\left(x+\;\ell \right)-\;v_{R}\;\left(x\right)$. Since $v_{R}$ is regular, it can be expanded as $v_{R}\left(x+\;\ell \right)-v_{R}\left(x\right)=\ell \nabla v_{R}$, so that $\nabla \cdot \tau^{\ell }∼\ell {\left(\nabla v_{R}\right)}^{2}=O\left(\ell \right)$.

Collecting the different term, we find that the l.h.s. of Equation (Equation 11) is the sum of the following orders: (12)$$\begin{array}{ccc} & O(\ell ) & :-\nabla \cdot \tau^{\ell } \end{array}$$(13)$$\begin{array}{ccc} & O(1) & :\partial_{t}{\bar{v_{R}}}^{\ell }+\nabla \cdot {\bar{v_{R}}}^{\ell }{\bar{v_{R}}}^{\ell }+\frac{\nabla {\bar{p_{R}}}^{\ell }}{\rho }-\nu \Delta {\bar{v_{R}}}^{\ell } \end{array}$$(14)$$\begin{array}{ccc} & O(1/\ell ) & :\dot{\gamma }\nabla_{\gamma }{\bar{v_{\alpha }}}^{\ell }+{\bar{\left(v_{\alpha }\cdot \nabla \right)v_{R}}}^{\ell } \end{array}$$(15)$$\begin{array}{ccc} & O(1/\ell^{2}) & :-{\dot{x}}_{\alpha }\nabla_{x}{\bar{v_{\alpha }}}^{\ell }+{\bar{\left(v_{R}\cdot \nabla \right)v_{\alpha }}}^{\ell } \end{array}$$(16)$$\begin{array}{ccc} & O(1/\ell^{3}) & :{\bar{\left(v_{\alpha }\cdot \nabla \right)v_{\alpha }}}^{\ell }+\frac{\nabla {\bar{p_{\alpha }}}^{\ell }}{\rho }-\nu \Delta {\bar{v_{\alpha }}}^{\ell } \end{array}$$

Cancelling the $O(1/\ell^{2})$ provides a first condition as: (17)$${\dot{x}}_{\alpha }\nabla_{x}{\bar{v_{\alpha }}}^{\ell }={\bar{\left(v_{R}\cdot \nabla \right)v_{\alpha }}}^{\ell }.$$
Due to the regularity of $v_{R}$, we can write ${\bar{\left(v_{R}\cdot \nabla \right)v_{\alpha }}}^{\ell }=v_{R}\left(x_{\alpha }\right)\cdot {\bar{\nabla v_{\alpha }}}^{\ell }$ for small enough *ℓ*. Condition (Equation 17) is then satisfied provided: (18)$${\dot{x}}_{\alpha }=v_{R}\left(x_{\alpha }\right).$$
Physically, this means that the singularity point is advected by the regular field surrounding it. Cancelling the O(1/ℓ) provides a second condition, as: (19)$$\dot{\gamma }{\bar{\nabla_{\gamma }v_{\alpha }}}^{\ell }=-{\bar{\left(v_{\alpha }\cdot \nabla \right)v_{R}}}^{\ell }.$$
Due to the regularity of $v_{R}$, we can write $({\bar{v_{\alpha }\cdot \nabla )v_{R}}}^{\ell }=\left({\bar{v_{\alpha }}}^{\ell }\cdot \nabla \right)v_{R}\left(x_{\alpha }\right)$. We then obtain the equation: (20)$$\dot{\gamma }{\bar{\nabla_{\gamma }v_{\alpha }}}^{\ell }=-\left({\bar{v_{\alpha }}}^{\ell }\cdot \nabla \right)v_{R}.$$
Physically, this means that the force axis and its direction are moved around by the shear of the regular field at the location of the singularity.

Cancelling the O(1) term provides the condition that ${\bar{v_{R}}}^{\ell }$ is a solution of the Navier–Stokes equation. Indeed, this is the idea behind the two fluid model, to allow for the scales above the coarse-graining to be described by a solution of the Navier–Stokes equation.

We are then left with the smallest order $O(\ell^{2})$ and the highest order term $O(\ell^{-3})$, which cannot be balanced in general. For the system to have a physical solution, we then impose a “bootstrap condition”, namely that the two terms must be of the same order of magnitude, thereby fixing the coarse-graining scale $\ell_{c}$ via $\nu^{2}/\ell_{c}^{3}∼\ell_{c}{\left(\nabla v_{R}\right)}^{2}$. We thus find $\ell_{c}={\left(\nu^{3}/\epsilon_{r}\right)}^{1/4}$, with $\epsilon_{r}=\nu {\left(\nabla v_{R}\right)}^{2}$ being the dissipation of the regular field. Therefore, the coarse-graining scale imposed by the bootstrap condition is precisely the Kolmogorov scale $\ell_{c}=\eta$.

Physically, this condition can be understood as follows: the singularity is dissipative, and to maintain it, one must apply a force. Such force is provided by the regular field, through the term $\nabla \cdot \tau^{\ell }$, which keeps track of the fraction of the velocity field that is sent to the subgrid scale, and which is taken into account by the pinçon. Conversely, the pinçon applies an extra turbulent stress onto the regular field that extends around it in a ball of radius of the order of the Kolmogorov scale. To keep a precise account of these effects, we thus split the Reynolds stress and the pinçon force into a contribution at $x_{\alpha }$ and a contribution around that location, and share the contribution among the pinçon, and the regular field.

Taking into account the fact that ${\bar{v_{R}}}^{\eta }\approx v_{R}$, ${\bar{v_{\alpha }}}^{\eta }=C_{\psi }\Pi_{\alpha }/\eta$ and $\dot{\gamma }{\bar{\nabla_{\gamma }v_{\alpha }}}^{\eta }\approx C_{\psi }\dot{\Pi_{\alpha }}/\eta$, the following system of equations to describe the coupling between the pinçon and the regular field is obtained:

for the regular field: (21)$$\begin{array}{ccc} \partial_{t}{\bar{v_{R}}}^{\eta }\left(x\right) & + & \left({\bar{v_{R}}}^{\eta }\cdot \nabla \right){\bar{v_{R}}}^{\eta }\left(x\right)+\frac{\nabla {\bar{p_{R}}}^{\eta }}{\rho }\left(x\right)-\nu \Delta {\bar{v_{R}}}^{\eta }\left(x\right)\\ \; & = & \nabla \cdot \tau^{\eta }\left(x\right)-\frac{\nu^{2}}{\eta^{3}}\psi \left(\frac{x-x_{\alpha }}{\eta }\right)F_{\alpha }; \end{array}$$

for the pinçon: (22)$$\begin{array}{ccc} {\dot{x}}_{\alpha } & = & {\bar{v_{R}}}^{\eta }\left(x_{\alpha }\right),\\ \dot{\Pi_{\alpha }} & = & -\left(\Pi_{\alpha }\cdot \nabla \right){\bar{v_{R}}}^{\eta }\left(x_{\alpha }\right)+\frac{\eta }{C_{\psi }}\nabla \cdot \tau^{\eta }\left(x_{\alpha }\right)-\frac{\nu^{2}\psi \left(0\right)}{C_{\psi }\eta^{2}}F_{\alpha }. \end{array}$$

These equations describe a two-fluid approach of turbulence, coupling a coarse-grained field at the Kolmogorov scale, and the pinçon with the Reynolds stress providing the necessary driving force to create pinçons. The latter are entities living below the Kolmogorov scale that are advected and sheared by the coarse-grained fluid, and that exert a forcing on the coarse-grained field which results in a dissipation of energy. To conclude our two-fluid model, we must prescribe the interactions between pinçons, in a way that is compatible with the Navier–Stokes equations.

### 2.5. Interactions of Pinçons

An ensemble of *N* pinçons, α=1,⋯,N produces a velocity field v(x,t):(23)$$v\left(x,t\right)=\sum_{\alpha }v_{\alpha }\left(x,t\right).$$

Around a pinçon α, the ensemble of other pinçons produces a regular field $v_{R}=\sum_{\beta \neq \alpha }v_{\beta }\left(x_{\alpha }\right)$. Motivated by such observation, we *define* the interactions of the pinçons via the following set of 2N differential equations: (24)$$\begin{array}{ccc} {\dot{x}}_{\alpha } & = & \sum_{\beta \neq \alpha }v_{\beta }\left(x_{\alpha },t\right), \end{array}$$(25)$$\begin{array}{ccc} {\dot{\Pi }}_{\alpha } & = & -\left(\Pi_{\alpha }\cdot \nabla_{x_{\alpha }}\right)\left(\sum_{\beta \neq \alpha }v_{\beta }\left(x_{\alpha },t\right)\right)-\frac{\nu^{2}\psi \left(0\right)}{C_{\psi }\eta^{2}}F_{\alpha }+\frac{E_{\alpha }}{C_{\psi }}\chi , \end{array}$$
where $E_{\alpha }=U_{\alpha }^{2}$ describes the local energy of the large scale regular field that provides a stochastic forcing χ via the (random) Reynolds stress contribution. In the sequel, we assume that χ is isotropic and shortly correlated over a Kolmogorov time scale, so that $\langle \chi_{i}\left(t\right)\chi_{j}\left(t^{\prime }\right)\rangle =\delta \left(t-t^{\prime }\right)\delta_{ij}$. Note that Equation (25) is a definition that leaves aside many conditions that may have to be satisfied for the model to be an exact representation of the small scales of Navier–Stokes flows. For example, this model is more likely to be valid as the dilute limit is achieved, so that the pinçons are sufficiently apart from each other for them to be considered as point-like particles. In addition, no distinction is made between close and distant interactions, while in the former case, diverging velocities and correlations may impede our possibility to consider that the field generated by the external fields is smooth enough so that the approximation ${\bar{\left(v_{R}\cdot \nabla \right)v_{\alpha }}}^{\ell }=v_{R}\left(x_{\alpha }\right)\cdot {\bar{\nabla v_{\alpha }}}^{\ell }$ is valid for small enough *ℓ*. Therefore, even if individually each pinçon is a weak solution of Navier–Stokes equations, the collection of *N* pinçon is not an exact weak solution of Navier–Stokes equations.

In some sense anyway, the equations of motions of the pinçons correspond to the equations that are imposed by the structure of the Navier–Stokes equations and the requirement that the local velocity field induced by each pinçon should obey such equations. Equations (Equation 24) and (Equation 25) can therefore be seen as the equivalent of the motion of poles or zeros of partial differential equations that have been computed, starting from Kruskal [27] for the KdV equations (see [28,29] for a review). The motions are furthermore constrained by the condition that they stay with the unit hypersphere such that $∥\gamma_{\alpha }∥<1$.

The pinçons are characterized by an interaction energy:(26)$$E=\frac{C_{\psi }}{\eta }\sum_{\beta \neq \alpha }\Pi_{\alpha }\cdot v_{\beta }\left(x_{\alpha },t\right).$$
Due to the presence of dissipation and forcing $F_{\alpha }$ and χ, this interaction energy is not conserved in general. However, there may exist situations where dissipation and forcing balance statistically, so that the system reaches an out-of-equilibrium steady state.

### 2.6. Weak Pinçon Limit

The equations of motions (Equation 24) and (Equation 25) take a simple expression, in the “weak pinçon” limit, where the intensity of the pinçons is very small, $\gamma_{\alpha }\ll 1$ for any α. In this case, $\Pi_{\alpha }=8\nu \gamma_{\alpha }/3$, and one can develop $\phi_{\alpha \beta }^{-1}=(1+\gamma_{\beta }\cdot r_{\alpha \beta }/∥r_{\alpha \beta }∥)$. The equations of motions under such approximations are: (27)$$\begin{array}{ccc} {\dot{x}}_{\alpha } & = & 2\nu \sum_{\beta \neq \alpha }\left[\frac{\gamma_{\beta }}{∥r_{\alpha \beta }∥}+\left(\gamma_{\beta }\cdot r_{\alpha \beta }\right)\frac{r_{\alpha \beta }}{∥r_{\alpha \beta }{∥}^{3}}\right], \end{array}$$(28)$$\begin{array}{ccc} {\dot{\gamma }}_{\alpha } & = & 2\nu \sum_{\beta \neq \alpha }\left[-\left(\gamma_{\alpha }\cdot \gamma_{\beta }\right)\frac{r_{\alpha \beta }}{∥r_{\alpha \beta }{∥}^{3}}+3\;\left(\gamma_{\alpha }\cdot r_{\alpha \beta }\right)\left(\gamma_{\beta }\cdot r_{\alpha \beta }\right)\frac{r_{\alpha \beta }}{\left|\right|r_{\alpha \beta }{\left|\right|}^{5}}\right.\\ & & \left.-\;\gamma_{\alpha }\frac{\gamma_{\beta }\cdot r_{\alpha \beta }}{∥r_{\alpha \beta }{∥}^{3}}+\gamma_{\beta }\frac{\gamma_{\alpha }\cdot r_{\alpha \beta }}{∥r_{\alpha \beta }{∥}^{3}}\right]-\frac{6\pi \nu \psi (0)}{C_{\psi }\eta^{2}}\gamma_{\alpha }+\frac{3E_{\alpha }}{8C_{\psi }\nu }\chi . \end{array}$$
These equations of motions are reminiscent of the equations of motions of the vortons (see Equation (Equation 48) in Appendix D), with vectorial products being replaced by a scalar product and additional terms appearing. However, the motion and intensities of the pinçons are driven by forces decaying, respectively, like 1/r and $1/r^{2}$, rather than respectively $1/r^{2}$ and $1/r^{3}$ for the vortons. Moreover, the pinçons are subject to a friction proportional to the viscosity.

The interaction energy in this case is:(29)$$E=\frac{16C_{\psi }\nu^{2}}{3\eta }\sum_{\beta \neq \alpha }\left[\frac{\gamma_{\alpha }\cdot \gamma_{\beta }}{∥r_{\alpha \beta }∥}+\frac{\left(\gamma_{\alpha }\cdot r_{\alpha \beta }\right)\left(\gamma_{\beta }\cdot r_{\alpha \beta }\right)}{∥r_{\alpha \beta }{∥}^{3}}\right],$$
which is the classical self interaction energy of pair of singularities [13].

## 3. Dynamics of a Pair of Pinçons

### 3.1. Interest of Considering a Pair of Pinçons

Previous experimental [30,31] and numerical investigations [32] about the location of the structures with extreme local energy transfer showed that they are located near interactions (possibly reconnection) of Burgers vortices. Previous and recent high-resolution numerical simulations of reconnection of anti-parallel vorticity filaments [33,34] showed that the process is associated with the formation of a local cusp over each filament that could possibly lead to a singular behaviour [35]. This possibility was confirmed by a detailed study of the interaction of two Burgers vortices conducted by [36,37,38,39,40] using the Biot–Savart approximation. They showed that the interaction indeed leads to a cusp formation on the vortex line, with very large, possibly diverging velocities at the tip of the cusp, with orientation along the bisector of the angle of the cusp. The reconnection was also found to be associated with a depletion of the helicity that eventually reaches zero at the reconnection [36]. In this picture, the associated pinçons created by a coarse-graining at the Kolmogorov scale would then be located at the tip of each cusp, with spins in the direction of the bisector of the cusps (see Figure 3a). This shows the interest of studying more closely the dynamics of a pair of pinçons and see how it compares with known features of the reconnection. This is the aim of the present section. In all the sequel, we renormalize the length by $r_{0}$, the distance between the two pinçons at time t=0 and the time by the associated viscous time $\tau_{\nu }=r_{0}^{2}/\nu$. We first start with the simplest case, where the pair constitutes a dipole.

### 3.2. Dynamics of a Dipole of Pinçons

#### 3.2.1. Equations

Let us consider the dynamics of a dipole, sketched in Figure 3b, made up of two pinçons located at $x_{\alpha }$ and $x_{\beta }$, and such that initially $\gamma_{\alpha }+\gamma_{\beta }=0$ and $x_{\alpha }-x_{\beta }=r_{0}r$.

We have then $v_{\beta }\left(x_{\alpha }\right)=-v_{\alpha }\left(x_{\beta }\right)\equiv \nu v\left(r\right)/r_{0}$ and $F_{\alpha }+F_{\beta }=0$. Using the aforementioned non-dimensionalization, we then get the equation of motion:(30)$$\begin{array}{ccc} {\dot{x}}_{\alpha } & = & \frac{\nu }{r_{0}}v,\\ {\dot{x}}_{\beta } & = & -\frac{\nu }{r_{0}}v,\\ {\dot{\Pi }}_{\alpha } & = & \Pi_{\alpha }\nabla_{r}V-\frac{\nu \psi (0)}{C_{\psi }}{\left(\frac{r_{0}}{\eta }\right)}^{2}F_{\alpha }+\frac{kT_{\alpha }r_{0}^{2}}{C_{\psi }\nu }\chi ,\\ {\dot{\Pi }}_{\beta } & = & -\Pi_{\beta }\nabla_{r}V-\frac{\nu \psi (0)}{C_{\psi }}{\left(\frac{r_{0}}{\eta }\right)}^{2}F_{\beta }+\frac{kT_{\beta }r_{0}^{2}}{C_{\psi }\nu }\chi . \end{array}$$
Therefore, the center of mass of the dipole $x_{\alpha }+x_{\beta }$ does not move, while the mean dipole strength $(\Pi_{\alpha }+\Pi_{\beta })/2$ obeys:(31)$${\dot{\Pi }}_{\alpha }+{\dot{\Pi }}_{\beta }=\frac{\left(E_{\alpha }+E_{\beta }\right)r_{0}^{2}}{2C_{\psi }\nu }\chi .$$
The forcing induces fluctuations proportional to the mean local energy $(E_{\alpha }+E_{\beta })/2$ that destroy the dipole geometry over a viscous time scale. It therefore only makes sense to study the dipole case in the low temperature limit where $(E_{\alpha }+E_{\beta })/2\to 0$.

#### 3.2.2. Results at Zero Temperature

Let us first investigate the dynamics in the zero temperature $E_{\alpha }+E_{\beta }=0$. In this case, the dipole remains exactly a dipole at all times, and we have $\Pi_{\alpha }=-\Pi_{\beta }\equiv \nu \Pi$, $F_{\alpha }=-F_{\beta }\equiv \;F$ and $\gamma_{\alpha }=\;-\;\gamma_{\beta }\equiv \;\gamma$. The dipole dynamics of the quantities characterizing the dipole, namely r and γ (or equivalently Π), can be obtained by taking the difference of the first two and the last two equations of Equation (Equation 30) to obtain: (32)$$\begin{array}{ccc} \dot{r} & = & 4\left(-\frac{\gamma +r/r}{r\phi_{*}}+\left(1-\gamma^{2}\right)\frac{r}{r^{2}\phi_{*}^{2}}\right), \end{array}$$(33)$$\begin{array}{ccc} \dot{\Pi } & = & \frac{2\Pi }{r^{3}\phi_{*}^{3}}A\left(\gamma ,\theta \right)\;r-\frac{\psi (0)}{C_{\psi }}{\left(\frac{r_{0}}{\eta }\right)}^{2}F, \end{array}$$
where
(34)$$\begin{array}{c} A\left(\gamma ,\theta \right)=\gamma (1-3\cos^{2}\left(\theta \right)-3\gamma \cos\left(\theta \right)-\gamma \cos^{3}\left(\theta \right)-2\gamma^{2}) \end{array}$$
where r=∥r∥, cos(θ)=(γ·r)/(rγ), $\phi_{*}=1+\gamma \cos\left(\theta \right)$ and Π=∥Π∥=G(γ).

Note that, from these expressions, the evolution of r and Π occurs in the plane generated by the two vectors r and γ. Thus, there remain only three independent quantities to determine the dipole axis and its orientation, namely *r*, θ and γ. The evolution of the first two quantities can be simply derived by projecting Equation (Equation 32) on $e_{r}$ and $e_{\theta }$, while the last quantity can be obtained by taking the scalar product of Equation (Equation 33) with Π to get an evolution for ${∥\Pi ∥}^{2}$, which leads to γ through $\gamma =G^{-1}\left(∥\Pi ∥\right)$. One thus obtains after straightforward simplifications: (35)$$\begin{array}{ccc} \dot{r} & = & \frac{4}{r}\left(\frac{1-\gamma^{2}}{\phi_{*}^{2}}-1\right), \end{array}$$(36)$$\begin{array}{ccc} r\dot{\theta } & = & 4\;\frac{\gamma \sin\theta }{r\phi_{*}}, \end{array}$$(37)$$\begin{array}{ccc} \dot{\Pi^{2}} & = & 4\frac{\Pi^{2}}{r^{2}}A\left(\gamma ,\theta \right)\cos\left(\theta \right)-\frac{2\psi (0)}{C_{\psi }}{\left(\frac{r_{0}}{\eta }\right)}^{2}F\cdot \Pi \end{array}$$
We have integrated the equations of motions (Equation 35) and (Equation 37) for fixed initial radius $r_{0}=1$ and $\gamma_{0}=0.1$ and various initial values of $\theta_{0}$ and taking $\psi \left(0\right)=1/{\left(2\pi \right)}^{3/2}$ (valid for ψ Gaussian). The resulting evolution is computed in two cases, with and without friction. The first case corresponds to the initial stage of the dynamics, just after the pinçons are created. Indeed, a pinçon is created with an initial force corresponding to the local Reynolds stress $F_{\alpha }=\eta^{3}/\left(\nu^{2}\psi \left(0\right)\right)\nabla \cdot \tau^{\eta }\left(x_{\alpha }\right)$, so that the dissipation is initially suppressed. In that case, we observe in Figure 4a that there are two fixed points of the dynamics for θ: one stable and attractive, corresponding to θ=π, and one unstable and repelling, corresponding to θ=0. As a result, the pinçons are mostly repelling each other, except when they start exactly anti-aligned and facing away each other, in which case they attract and annihiliate each other. The resulting dynamics can also be appreciated in the phase space, as shown in Figure 5a.

Interesting scaling laws are observed in the two stages that are reminiscent of what is observed during reconnection events. The pinçons with initial inclination in the interval [0,π/2] and different from 0 start moving towards each other, while decreasing their strength and increasing their angle, in absolute value. Once they reach the value θ=π/2, they change direction and get away from each other (see Figure 4a). During the collapse stage, the radius of the dipole decreases approximately like $\sqrt{t_{c}-t}$, which is the Leray scaling [19]. The collapse stage is nearly universal, with weak dependence on the initial angle, while the escape depends more strongly on the initial orientation. Such asymmetry has also been observed in reconnection of quantum vortices [41]. During the separating stage, θ gets closer to π and there is also an approximate power law escape law $r∼\sqrt{t-t_{c}}$. The scaling laws are explored further in the general case in Section 3.4.

At a later stage, the initial Reynolds stress has decayed and cannot balance the pinçons dissipation anymore. The extreme case is when the Reynolds stress has decayed to zero, and when the dissipation is the strongest. This case has been studied by running another set of simulations including the friction, starting the dipole at $r_{0}=\eta$. It is shown in Figure 4b. We notice that, as expected, the dissipation now induces a constant decline of the dipole intensity that eventually results in the death of the dipole. The resulting phase space is shown in Figure 5b, where a plunging funnel corresponding to dissipation can now clearly be seen. There are now two situations depending on whether the initial angle is less than or bigger than π/2. In the first case, the dipole contracts and orientates itself towards π/2 before its death. This situation can be associated with a reconnection event. For initial angles greater than π/2, the dipole expands while keeping its initial orientation before eventually dying and stopping.

In summary, the natural dynamics of a dipole without stochastic forcing is always dissipative, resulting in the final death of the dipole. Before its death, the dipole can experience either a first contraction stage of its initial angle is less than π/2, with dynamics resembling reconnection events, or experiences and expansion while it tries to anti-align its two components (θ→π). The question is now whether the dipole can be maintained for a longer time, and maintain another orientation if we take the stochastic forcing into account. As we showed in the beginning, as soon as we add some stochastic noise, the dipolar geometry breaks down on a viscous time scale. There is therefore a subtle interplay to be understood between the decay of the dipole, the forcing and the departure from dipolar geometry. There is, however, an interesting observation that allows for tackling the problem in a simple and elegant manner. Indeed, due to the friction, the pinçon intensity decreases with time, and we are likely to enter into the weak pinçon limit after a sufficient long time. The latter is much simpler to code and understand, since it resembles the dynamics of dipolar moments. In the sequel, we therefore investigate the finite but small forcing limit in the weak pinçon approximation.

### 3.3. Results at Finite Small Temperature

#### 3.3.1. From Pair of Pinçons to Dipole Equations in the Weak Pinçon Limit

In such limit, the dipole dynamics and the evolution of the dipole radius, orientation and strength can be obtained from the evolution of three quantities: $R=r^{2}$, C=γ·r and $N=\gamma^{2}$. To do so, Equations (Equation 27) and (Equation 28) are first considered for two pinçons $\gamma_{\alpha }$ and $\gamma_{\beta }$, and three quantities are introduced: $r=x_{\alpha }-x_{\beta }$, $\Gamma =(\gamma_{\alpha }+\gamma_{\beta })/2$ and $\Pi =(\gamma_{\alpha }-\gamma_{\beta })/2$. By combination of the equations of motions, one get the three coupled equations:(38)$$\begin{array}{ccc} \dot{r} & = & -4\nu \left(\frac{\Pi }{r}+r\frac{\Pi \cdot r}{r^{3}}\right),\\ \dot{\Pi } & = & 2\nu \left[\left(\Pi^{2}-\Gamma^{2}\right)\frac{r}{r^{3}}+3\frac{r}{r^{5}}\left({\left(\Gamma \cdot r\right)}^{2}-{\left(\Pi \cdot r\right)}^{2}\right)\right]-\frac{6\pi \nu \psi (0)}{C_{\psi }\eta^{2}}\Pi +\frac{3(E_{\alpha }-E_{\beta })}{16C_{\psi }\nu }\chi ,\\ \dot{\Gamma } & = & 4\nu \left[\Gamma \frac{\left(\Pi \cdot r\right)}{r^{3}}-\Pi \frac{\left(\Gamma \cdot r\right)}{r^{3}}\right]-\frac{6\pi \nu \psi (0)}{C_{\psi }\eta^{2}}\Gamma +\frac{3(E_{\alpha }+E_{\beta })}{16C_{\psi }\nu }\chi . \end{array}$$
The last equation of Equation (Equation 38) shows that Γ is forced by $\bar{E}=\left(E_{\alpha }+E_{\beta }\right)/2$. If we start with a dipole condition Γ=0, in the small temperature limit $\bar{E}\ll 1$ and for a time scale that is short with respect to the diffusive time scale $\tau_{\nu }=r_{0}^{2}/\nu$, all the terms proportional to Γ in the second equation of Equation (Equation 38) can be ignored. Then, the first equation is multiplied by r and the second equation by Π to obtain equations for *R* and *N*, and the first equation multiplied by Π is summed with the second equation multiplied by r to get the equation of evolution for *C*. After non-dimensionalization by $r_{0}$ and $r_{0}^{2}/\nu$ and rearrangement, one then obtains the three coupled equations:(39)$$\begin{array}{ccc} \dot{R} & = & -16\frac{C}{R^{1/2}},\\ \dot{C} & = & -2\left[\frac{N}{R^{1/2}}+5\frac{C^{2}}{R^{3/2}}\right]-\rho C+R^{1/2}\mu \zeta ,\\ \dot{N} & = & 4\left[N\frac{C}{R^{3/2}}-3\frac{C^{3}}{R^{5/2}}\right]-\rho N+N^{1/2}\mu \zeta , \end{array}$$
where ζ is a delta correlated white noise obeying $<\zeta \left(t\right)\zeta \left(t^{\prime }\right)>=\delta \left(t-t^{\prime }\right)$, and the friction ρ and forcing μ coefficient are given by
(40)$$\begin{array}{ccc} \mu & = & \frac{3r_{0}^{2}\left(E_{\alpha }-E_{\beta }\right)}{16\nu^{2}C_{\psi }},\\ \rho & = & 6\pi \psi \left(0\right)\frac{r_{0}^{2}}{\eta^{2}C_{\psi }}. \end{array}$$
From *N*, *R* and *C*, we then obtain $r=R^{1/2}$, $\gamma =N^{1/2}$ and θ=arccos(C/(rγ)).

To check the validity of the weak pinçon approximation, the comparison between the full dynamics computed from Equation (Equation 33), and its weak limit Equation (Equation 40) in the zero temperature limit μ=0 and without (ρ=0) and with friction is shown in Figure 6. We see that the two dynamics coincide very well for most cases, and that the approximation is better on the late stage, when there is friction. Indeed, the friction forces decay of the dipole intensity.

The weak dipole limit actually helps us identifying special angles for the dipole dynamics. Indeed, from the last equation of Equation (Equation 40), the first term of the r.h.s cancels whenever $\xi \left(\theta \right)=\cos\left(\theta \right)\left(1-3\cos^{2}\left(\theta \right)\right)=0$, corresponding to the three angles in the interval [0,π], namely θ=π/2, $\theta =\arccos(1/\sqrt{\left(3\right)})$ and $\theta =\pi -\arccos(1/\sqrt{\left(3\right)})$. Those angles are identified by black dotted lines in Figure 4 and Figure 6. When the forcing and dissipation vanish or balance, they correspond to special directions where the dipole intensity can remain stationary. Indeed, the angle θ=π/2 partitions the dynamics since the angles θ(t) increase and the radius decreases if and only if θ(t) is smaller than $\frac{\pi }{2}$. The two other angles do not seem to play a specific role in the zero temperature limit. However, the situation is different in other situations, at a finite temperature, as shown in the following section.

#### 3.3.2. Dynamics at Finite Temperature

Integrating (Equation 40) allows for efficiently studying the dynamics of a noisy dipole, provided two criteria are satisfied: (i) γ≤0.5, in order to be in the weak limit and (ii) *r* is large enough ($r\geq \frac{1}{\sqrt{\rho }}$) so that the two pinçons may still be considered as distinct. Moreover, for numerical reasons, we stop the integration whenever γ≤0.01 or $r\geq \sqrt{\rho }$, in which case we consider that the pinçons are either dying or that the dipole has escaped to infinity. In practice, most of our simulations were stopped because either γ≤0.01 or γ≥0.5. Figure 7 shows the evolution of a dipole satisfying the equations of motion (Equation 40) for fixed initial radius $r_{0}=1$ and $\gamma_{0}=0.1$, various initial values of $\theta_{0}$ and ρ=0.12 and μ=0.009. Several tendencies emerge, as illustrated in Figure 7. First, the integration time is slightly longer as the noise may remain γ stationary for a certain amount of time. Second, the evolution of θ(t) is not monotonic anymore and the angle θ tends to get closer to the values cancelling ξ(θ), and tend to θ=π/2 in most of the cases. As a result, the dipole can be maintained for some time in a non-equilibrium balance, where dissipation exactly balances the stochastic forcing, allowing the dipole intensity to decay less rapidly, and the dipole to live longer. We have checked that, when the noise is too large, then fluctuations are enough to bring the dipole intensity close to the limit γ=0 or γ=1 in a finite time, resulting in the dipole collapse or death quicker than in the zero temperature limit.

### 3.4. General Dynamics of a Pair of Pinçons

#### 3.4.1. Short Time Dynamics

In the case of dipole, the short time dynamics correspond either to an escape with θ→π and γ→1 or first a contraction (close interaction) and then an escape. Here, we consider the general case of a pair of pinçons to determine whether such observation is robust or not. The relative dynamics are characterized in this case by six independent scalar variables which are the distance *r* between the pinçons, their intensities $\gamma_{\alpha }$ and $\gamma_{\beta }$, their angles $\theta_{\alpha }$ and $\theta_{\beta }$ defined by $\cos\;\theta_{\alpha }=\left(\gamma_{\alpha }\cdot r_{\alpha \beta }\right)/\left(r\gamma_{\alpha }\right)$ and $\cos\;\theta_{\beta }=\left(\gamma_{\beta }\cdot r_{\beta \alpha }\right)/\left(r\gamma_{\beta }\right)$ and the angle φ defined by $\cos\;\varphi =\left(\gamma_{\alpha }\cdot \gamma_{\beta }\right)/\left(\gamma_{\alpha }\gamma_{\beta }\right)$. The dipole case studied in the previous section corresponds to $\gamma =\gamma_{\alpha }=\gamma_{\beta }$, $\theta =\theta_{\alpha }=\theta_{\beta }$ and φ=π. We then integrate the dynamics with Equations (Equation 24) and (Equation 25) without dissipation or noise for different initial conditions of these parameters, except that we always set $r_{0}=1$. Indeed, since the pinçon are created with an initial force corresponding to the local Reynolds stress $F_{\alpha }=\eta^{3}/\left(\nu^{2}\psi \left(0\right)\right)\nabla \cdot \tau^{\eta }\left(x_{\alpha }\right)$, the short time dynamics correspond to the case without dissipation or forcing.

Because there is no dissipation, we expect γ to tend toward 1 as for the dipole, so we implement a stopping condition when there is one pinçon for which γ>1−ε, with epsilon a small parameter taken here to $\varepsilon =10^{-2}$. We have run the dipole dynamics with many different initial conditions, and identified three scenarios:(i)repelling dipolar expansion, illustrated in Figure 8a. This case corresponds to the case where the two components run away from each other and gradually become a repelling dipole: their mutual angle ϕ tends to π, while they become anti-parallel to their separation vector θ→π and their intensities become equal to each other and tend to 1. In this case, the role of each pinçon is symmetric.(ii)aligned expansion, illustrated in Figure 8b. In this case, one component grows larger than the other one, while both pinçons become aligned with their separation vector and point in the same direction $\theta_{1}=\pi$, $\theta_{2}=0$, ϕ=0. The component with the lower intensity moves faster and speeds ahead of the other one.(iii)explosive collapse, illustrated in Figure 9a. In this case, the two pinçons are attracted to each other, while one of the two pinçons rapidly reaches the asymptotic value γ=1, corresponding to an infinite dissipation. In contrast with the expansion situations where the pinçons tend to align or anti-align, the collapse case corresponds to intermediate values of φ and θ different from π. This case can therefore be considered as the generic reconnection event.

**Figure 8 entropy-24-00897-f008:**
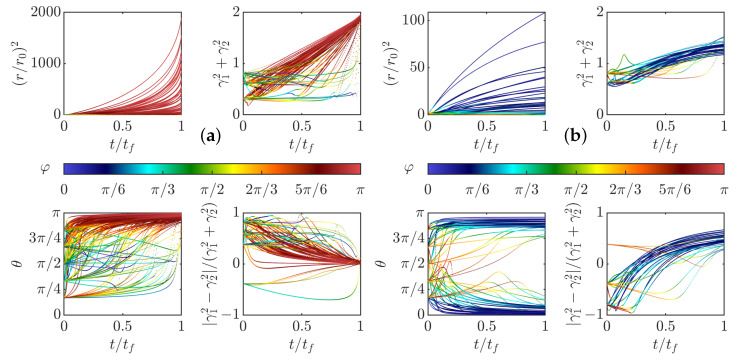
Time evolution of the distance *r*, the pinçons angles $\theta_{1}$, and $\theta_{2}$ (one line for each), total intensity $\gamma_{1}^{2}+\gamma_{2}^{2}$ and anisotropy $\left(\gamma_{1}^{2}-\gamma_{2}^{2}\right)/\left(\gamma_{1}^{2}+\gamma_{2}^{2}\right)$ in the two expansion cases. (**a**) case of repelling dipolar expansion; (**b**) case of aligned expansion. The points are colored by the value of the pair mutual angle, φ.

**Figure 9 entropy-24-00897-f009:**
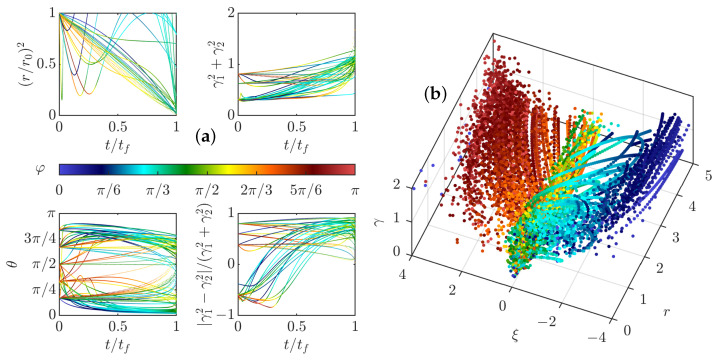
(**a**) Time evolution of the distance *r*, the pinçons angles $\theta_{1}$, and $\theta_{2}$ (one line for each), total intensity $\gamma_{1}^{2}+\gamma_{2}^{2}$ and anisotropy $\left(\gamma_{1}^{2}-\gamma_{2}^{2}\right)/\left(\gamma_{1}^{2}+\gamma_{2}^{2}\right)$ in the case of explosive collapse. (**b**) Phase space for a pair of pinçons in the initial stage of the dynamics, just after pinçons creation. The phase space is *r*, $\gamma =\gamma_{1}^{2}+\gamma_{2}^{2}$ and $\xi =\xi \left(\theta_{1}\right)\xi \left(\theta_{2}\right)$. The points are colored by the value of the pair mutual angle φ in all the figures.

Interestingly, the different cases partition in different areas the phase space *r*, $\gamma =\gamma_{1}^{2}+\gamma_{2}^{2}$ and $\xi =\xi \left(\theta_{1}\right)\xi \left(\theta_{2}\right)$, with $\xi \left(\theta \right)=\cos\left(\theta \right)\left(1-3\cos^{2}\left(\theta \right)\right)$, as illustrated in Figure 9b. One sees that the collapse mode tends to occur around ξ=0 meaning that at least one of the two pinçons tend to orientate at π/2, $\arccos(1/\sqrt{3})$ or $\pi -\arccos(1/\sqrt{3})$ from the separation vector. In contrast, the two expansion modes proceed with ξ=±2, corresponding to situations where the pinçons are aligned or anti-aligned with their separation vector. Summarizing, two new cases are found with respect to the dipole dynamics, namely a new mode of expansion, made with two aligned pinçons following each other, and a new mode of collapse following the law $\xi \left(\theta_{1}\right)\xi \left(\theta_{2}\right)=0$, with one component reaching the asymptotic value γ=1, and corresponding to a generic reconnection event. In the sequel, we study in more details these events.

#### 3.4.2. Scaling Laws of Collapse

During the collapse stage, the radius evolution is fitted with a power law in order to compare the exponent with the Leray scaling $\sqrt{t_{c}-t}$ [19]. Figure 10a shows the histogram of the values of the exponent obtained by fitting the law $r\left(t\right)=\beta_{c}{\left(t_{c}-t\right)}^{\delta }$ for the collapse cases. We observe that most of the values are near 1/2 which correspond to the Leray exponent. Figure 10b shows the time dynamics of rescaled squared radius. The Leray scaling is found to be verified asymptotically as $t/t_{c}\to 1$.

#### 3.4.3. Full Collapse Dynamics

The full dynamics of reconnection events can be investigated using a patching between the short time behavior, and the large time behavior, allowing e.g., the turbulent stress to decay like $\exp(-t/\tau_{\mathrm{forcing}})$, where $\tau_{\mathrm{forcing}}$ is a time scale associated with large scales. As shown in Figure 11, for the same initial configuration, when dissipation is high and the forcing characteristic time is short, the pinçons die very fast with no close interaction. On the contrary, if the dissipation is too low with a forcing persistent enough, the dynamics are very similar to the case without dissipation, and the pinçons still collapse in an explosive manner with their intensities tending to 1. Two intermediate cases are found where we have a collapse stage followed by a separation stage without explosion. These typical examples are illustrated in Figure 12a,b. In both cases, we observe first a collapse phase and then a separation phase although the particular dynamics are quite different. In the case of (a) where the dissipation and the forcing characteristic time are rather small, the transition between the two phases happens at a closer distance and has a configuration similar to the dipole with one of the pinçons axis abruptly turning from an angle close to 0 to an angle close to π, then the pinçons die very fast. In the case of (b) with larger dissipation and a more persistent forcing, the dynamics are smoother with hardly any change in the dipole relative orientation, only the axis angles change slowly and the pinçons survive a long time with a stable configuration. During the interaction, the maximum velocity and vorticity near the pair of pinçons, shown in Figure 13, exhibit marked oscillations due to the finite resolution of the grid. Using a moving average, we see, however, that for the cases where the intensities tend to 1, the maximum velocity and vorticity tend to infinity as expected. If we now look at the case of close interaction with a final separation corresponding to Figure 12a, they first decrease during the collapse stage until the time of minimum of *r*, after which they increase until the angle is close to ϕ=π; then, they finally both decay to zero when separating. This behaviour is reminiscent of what is happening during a reconnection of vortex rings, where the distance between rings decay like $\sqrt{t_{c}-t}$, with maximum velocity and vorticity growing up and then decaying [35].

## 4. Discussion

We have introduced a model of singularities of Navier–Stokes, named pinçonsthat are discrete particles characterized by their position and “spin”. These particles follow a nontrivial dynamics, obtained by the condition that the coarse-grained velocity field around these singularities obeys locally Navier–Stokes equations. We have shown that this condition can only be satisfied provided the coarse-graining scale is of the order of the Kolmogorov scale. When immersed in a regular field, the pinçons are further transported and sheared by the regular field, experiencing a friction together with an energy injection coming from by Reynolds stress of the regular field. We have used these properties to study the interaction of two pinçons, at the early and late stage of their evolution, and in the presence or absence of a stochastic forcing induced by the possible Reynolds stress.

Quite interestingly, we have identified several modes of interactions at short times that are characterized by the values of the parameter $\xi \left(\theta \right)=\cos\left(\theta \right)\left(1-3\cos^{2}\left(\theta \right)\right)$, where θ is the angle between the spin of the pinçon and the axis of the pair. Specifically, in the absence of noise, we identified two modes of expansion of the pair with ξ=±2, corresponding to situations where the pinçons are aligned or anti-aligned with their separation vectors, and one mode of collapse with ξ=0. In the presence of noise, we observe and additional transient non-equilibrium steady state expansion mode, with ξ=0, and the pinçons are perpendicular to the axis of the pair. The quantity $1-3\cos^{2}\left(\theta \right)$ actually plays an important role in the theory of liquid crystals, as its average defines the order parameter of the system $s=(1-3\cos^{2}\left(\theta \right))/2$, with possible transitions between liquid (s=0) and nematic phase (s=1). The different interaction modes therefore open the way to interesting different collective behaviors when considering a larger number of pinçons. Whether such behaviors are of relevance to the actual physics of turbulence is still an open issue, as the pinçon model ignores on a number of issues that may limit its range of validity: existence of large nonlocal energy transfer at the Kolmogorov scale, dilute approximation for the pinçon, scale separation between the pinçon and the ambient large scale velocity field, to name but a few.

Our study of the interaction of two pinçons, however, already revealed some interesting similarities with reconnection between two vortex rings. Indeed, we observe that the collapse generally obeys the $\sqrt{t_{c}-t}$ scaling that is observed during reconnection, and is characterized by transient growth of velocity or vorticity like in the reconnection. From another point of view, the pinçons dynamics are also reminiscent of the two-fluid model of superfluid, where the “regular” field, made of phonons, interact with the local topological defects that form the quantized vortices. Indeed, as shown by [41], the interaction of quantized vortices leads to Leray scaling, with distance between vortices decaying like $\sqrt{t_{c}-t}$. In a broader sense, the description of the interaction between pinçons and a regular field is parallel to the interaction of localized wave packets interacting with a mean flow, in the WKB-RDT model of [42]. By analogy, one may then wonder whether it would be possible to use the pinçons as a subgrid scale model of turbulence, allowing for describing the interaction of a velocity field filtered at the Kolmogorov length, with a collection of pinçons that encode the very intense energy transfers that are observed when scanning very small scales of turbulence [30]. If the pinçon model proved accurate enough to describe small scale turbulence, it would then enable the use of larger time-steps, as the motion of the small scale motions is governed by Lagrangian motions. Another issue is whether a short range regularization is needed at short distances to make the model applicable to subgrid modelling.

More generally, the pinçon model shares some interesting properties with other discrete models found in fluid mechanics, such as the vorton model of Novikov [10], derived from the 3D Euler equations, or the point-vortex model, derived from the 2D Euler equation: like them, it is a N-body model, describing the motion and “momentum” of entities with long range interactions. At variance with them, however, the pinçon model is not Hamiltonian because pinçons only exist in the presence of viscosity, so that energy is not conserved (and is actually dissipated locally by the singularity). To maintain pinçon, we thus have to introduce a large scale that provides constantly energy to the system (in our case via stochastic forcing). Both vorton model and point-vortex model are contributing to the progress of our understanding of turbulence phenomenology by providing simple toy models to play with: for example, the notion of “negative temperature” introduced by Onsager to explain the inverse cascade of turbulence, after he observed a similar process occurring in a point-vortex model.

We can then think of all these discrete models as “Ising models” of turbulence that can be used to get insight on turbulence properties. Therefore, even if the pinçon model does not accurately describe the behavior of small scale turbulence, it is an out-of-equilibrium statistical model of Navier–Stokes singularities with many interaction modes that bears some similarity with liquid crystal interactions. It may then stimulate new ideas regarding turbulence dynamics and properties and play a similar role than the Ising model in statistical mechanics.

## Figures and Tables

**Figure 2 entropy-24-00897-f002:**
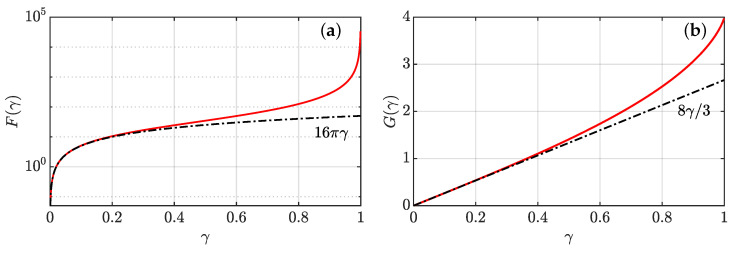
Parameters of a pinçon as a function of its intensity γ. (**a**) intensity of the force produced by the pinçon at its location. The black dashed line has equation y=16πγ; (**b**) generalized momentum of a pinçon. The black dashed line has equation y=8γ/3.

**Figure 3 entropy-24-00897-f003:**
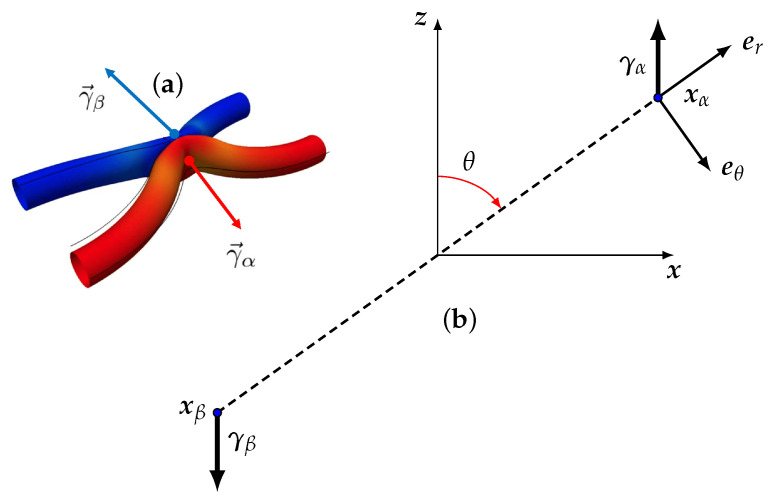
(**a**) Schematic geometry of pinçons creation at reconnection; (**b**) geometry of the dipole: two pinçons located at $x_{\alpha }$ and $x_{\beta }$, and such that initially $\gamma_{\alpha }+\gamma_{\beta }=0$. By convention, the angle θ is the angle between $\gamma_{\alpha }$ and $r=x_{\alpha }-x_{\beta }$. The (**a**) is adapted from Figure 3 of [34].

**Figure 4 entropy-24-00897-f004:**
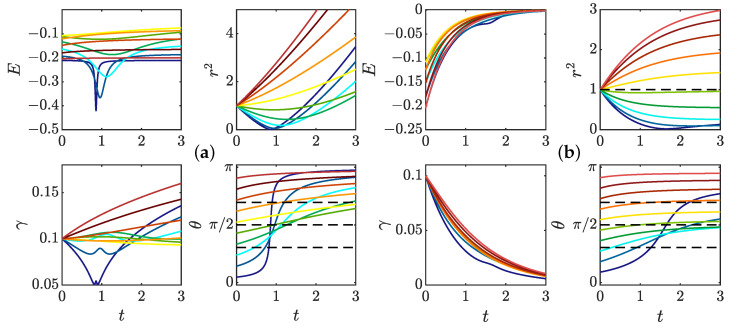
Dynamics of a dipole of pinçon for various initial conditions and (**a**) without friction, corresponding to the initial stage of the dynamics, just after pinçons creation; (**b**) with friction coefficient 0.7, corresponding to the late stage dynamics. The radius is initially fixed to r=1, the dipole intensity is initially set to γ=0.1 and the initial dipole orientation is fixed at different values between 0 and π (identified by different colors). The panel represents the time evolution of the different quantities: *E*: Interaction energy; $r^{2}$: Square of Dipole separation; γ: Dipole intensity; θ: Dipole orientation. Black dashed lines on θ(t) figure correspond to $\arccos\left(\frac{1}{\sqrt{3}}\right),\frac{\pi }{2}$, and $\pi -\arccos\left(\frac{1}{\sqrt{3}}\right)$.

**Figure 5 entropy-24-00897-f005:**
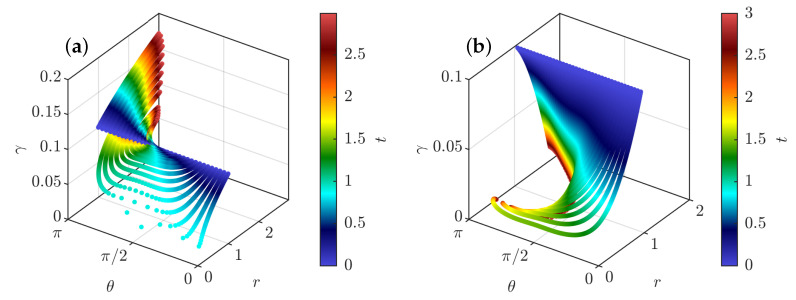
Phase space for a dipole of pinçons color-coded by the time, shown on the color bar. (**a**) Without friction, corresponding to the initial stage of the dynamics, just after pinçons creation; (**b**) With friction—coefficient 0.7—corresponding to the late stage dynamics. The radius is initially fixed to r=1, the dipole intensity is initially set to γ=0.1 and the initial dipole orientation is fixed at different values between 0 and π.

**Figure 6 entropy-24-00897-f006:**
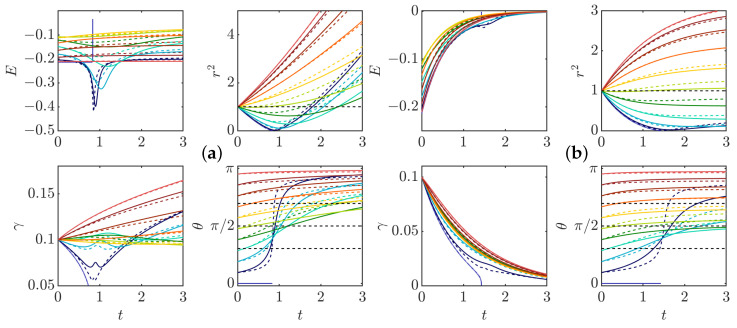
Comparison with the weak limit approximation: solid lines: complete model; dashed line: weak limit; (**a**) without friction; (**b**) with friction, coefficient 0.7. The panel represents the time evolution of the different quantities: *E*: Interaction energy; $r^{2}$: Square of Dipole separation; γ: Dipole intensity; θ: Dipole orientation. Black dashed lines on θ(t) figure correspond to $\arccos\left(\frac{1}{\sqrt{3}}\right),\frac{\pi }{2}$, and $\pi -\arccos\left(\frac{1}{\sqrt{3}}\right)$.

**Figure 7 entropy-24-00897-f007:**
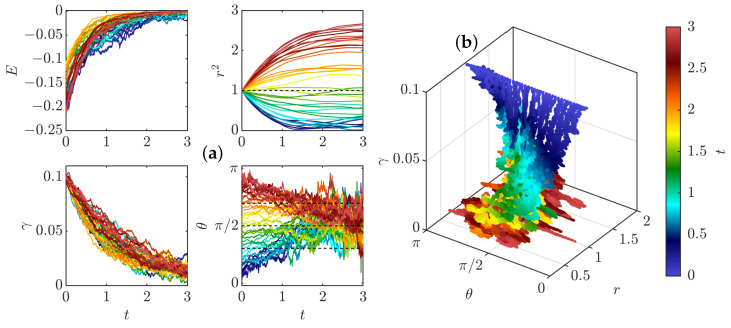
Effect of noise in the weak limit dissipative case for dissipation coefficient 0.7 (**a**) on the dynamics. The panel represents the time evolution of the different quantities: *E*:Interaction energy; $r^{2}$ Square of Dipole separation; γ: Dipole intensity; θ: Dipole orientation. Black dashed lines on a θ(t) figure correspond to $\arccos\left(\frac{1}{\sqrt{3}}\right),\frac{\pi }{2}$, and $\pi -\arccos\left(\frac{1}{\sqrt{3}}\right)$; (**b**) on the phase-space, color-coded by the time, shown on the color bar.The radius is initially fixed to r=1, the dipole intensity is initially set to γ=0.1 and the initial dipole orientation is fixed at different values between 0 and π. The intensity of the noise is μ=0.009.

**Figure 10 entropy-24-00897-f010:**
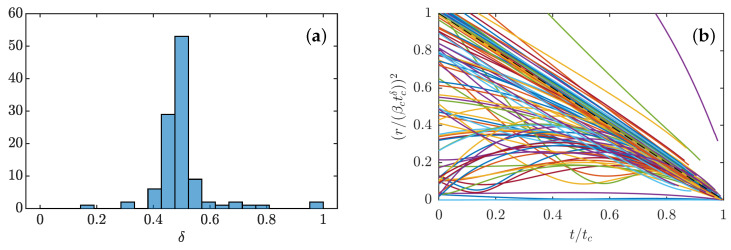
(**a**) Histogram of the values of the power law exponent δ; (**b**) squared distance rescaled by for the explosive collapse cases. The black dashed line corresponds to the Leray scaling with δ=1/2. We see that, for $t/t_{c}$ close to 1, most of the curves follow a power law with a power exponent δ close to 1/2.

**Figure 11 entropy-24-00897-f011:**
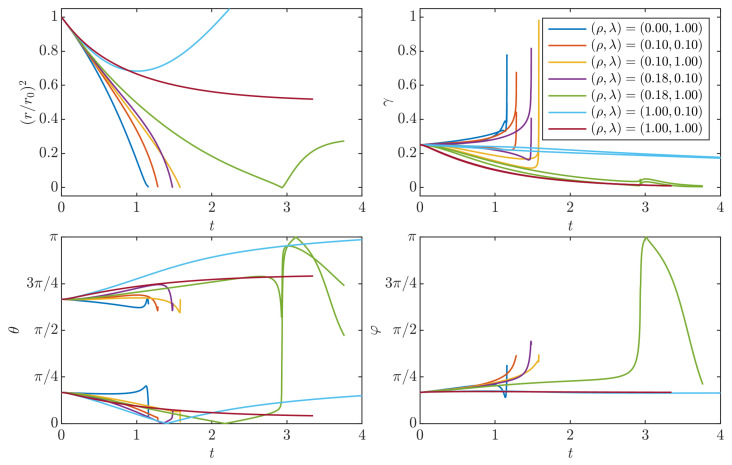
Time evolution of the different variables characterizing the pair of pinçons for seven cases with the same initial configuration but with different forcing characteristic time coefficient $\lambda =\tau_{\nu }/\tau_{\mathrm{forcing}}=r_{0}^{2}/\left(\nu \tau_{\mathrm{forcing}}\right)$ and dissipation coefficient $\rho =\left(\psi \left(0\right)/C_{\psi }\right){\left(r_{0}/\eta \right)}^{2}$. We see that, on the one hand, when dissipation is high and the forcing time is short, the pinçons die very fast with no close interaction. On the other hand, if the dissipation is low, the dynamics are very similar to the case without dissipation, and the pinçons still collapse in an explosive manner with their intensities tending to 1. Two intermediate cases are found where we have both the collapse dynamics and a separation dynamics without explosion.

**Figure 12 entropy-24-00897-f012:**
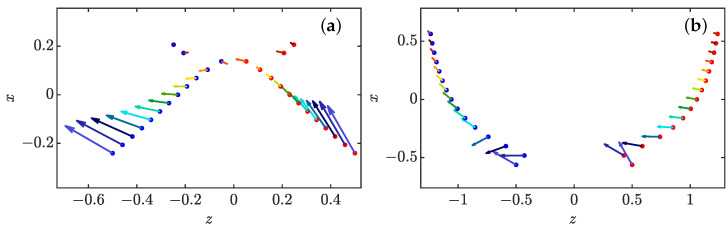
Dynamics of a pair of pinçons as a function of time in the plane defined by $\gamma_{1}$ and $r=re_{z}$ for two different forcing characteristic time coefficient (λ) and dissipation coefficient ρ: (**a**) (ρ,λ)=(0.18,1.00) and (**b**) (ρ,λ)=(1.00,0.10). The vectors gives the projection of $\gamma_{1}$ and $\gamma_{2}$ in the plane and the color codes the time, from t=0 (dark blue) to $t=t_{\mathrm{final}}$ (dark red), as well as the coordinate of the points on the vertical axis *x*. In both cases, the two pinçons (blue and red points) move initially towards each other (the distance is read on the horizonal axis *z*) and then their orientations change and they repel each other. In the case of (**a**), the transition between the two phases has a configuration similar to the dipole with one of the pinçons axis abruptly turning from an angle close to 0 to an angle close to π, then the pinçons die very fast. In the case of (**b**), the dynamics are smoother with hardly any change in the dipole relative orientation, only the axis angles change slowly, and the pinçons survive a long time with a stable configuration.

**Figure 13 entropy-24-00897-f013:**
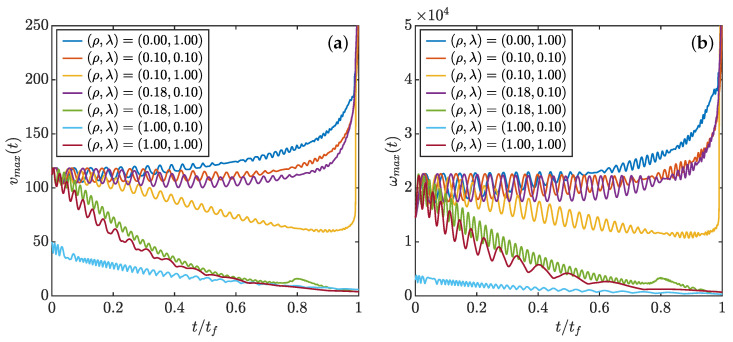
Time evolution of the maximum of rescaled velocity field (**a**) and vorticity field (**b**) around a pair of pinçons with same initial configurations and different forcing characteristic time coefficient $\lambda =\tau_{\nu }/\tau_{\mathrm{forcing}}$ and dissipation coefficient ρ. The rescaling is the same as in Figure 1, with $r_{0}$ the initial separation between the pinçons. The values correspond to a moving average over 15 time steps.

**Table 1 entropy-24-00897-t001:** Order of the various terms appearing in Equation (Equation 11) as a function of the filter length *ℓ* in the limit $x\to x_{\alpha }$.

$v_{\alpha }$	$\dot{\gamma_{\alpha }}\partial_{\gamma }\left(v_{\alpha }\right)$	${\dot{x}}_{\alpha }\partial_{x}v_{\alpha }$	$\left(v_{\alpha }\cdot \nabla \right)v_{R}$	$\left(v_{R}\cdot \nabla \right)v_{\alpha }$	$\nabla \cdot {\bar{v_{R}}}^{\ell }{\bar{v_{R}}}^{\ell }$
1/ℓ	1/ℓ	$1/\ell^{2}$	1/ℓ	$1/\ell^{2}$	1

## Data Availability

Codes used in this work are available at doi:10.5061/dryad.8sf7m0cqz (accessed on 23 June 2022).

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
