# Peer review of "A Model of Interacting Navier–Stokes Singularities"

_entropy, 2022, doi:10.3390/e24070897_

Round 1

Reviewer 1 Report

Report on: "A model of interacting Navier-Stokes singularities" by: Hugues Faller, Lucas Fery, Damien Geneste and Bérengère Dubrulle.

This is a nice theoretical and numerical paper dealing with a new point vortex 3D model of local quasi-singularities in Navier-Stokes flows. The physical motivations of the work are clearly expressed and there is a rather complete review of previous work on the subject. The new results are discussed and put into perspective. 

I think the paper is worthy of publication in Entropy as is, however I would like the authors to considered the few following remarks.

It is tempting to wonder what the equivalent of a Navier-Stokes pinçon would be in the Gross-Pitaevskii description of zero temperature superfluid. Do the authors have any idea on this subject?

Also, there is some previous work on ring singularities in Navier-Stokes that the authors might want to mention (they do not have to if they do not feel that the reference is relevant): 

K. B. Ranger, "A nonlinear ring singularity for the motion of a viscous incompressible liquid," in Quarterly Journal of Mechanics and Applied Mathematics, vol. 54, no. 1, pp. 57-64, Feb. 2001, doi: 10.1093/qjmam/54.1.57.

Author Response

We thank the reviewer for its useful comments and interesting question.
Below are point-to-point answers.

  • It has been proven theoretically that Gross-Pitaevskii equation is regular, owing to the dispersive term. Therefore, there is unfortunately no equivalent of the pinçons for this equation (please remember that the quantum vortices in GP are singularity of the Madelung transformation, not of the equation itself!)

  • We thank the reviewer for this interesting reference, but it did not appear relevant to us in the present context.

Reviewer 2 Report

Review article entitled: A model of interacting Navier-Stokes singularities

The article is very well described. And in my opinion, in its present form, it can be published. However, I have a few editorial comments:

1.      Please use the passive voice. In my opinion, the active voice is used too much.

2.      What is the practical application of the model cited? Are these just theoretical divagations or can they be applied as an alternative/additive turbulence model?

Author Response

We thank the reviewer for its useful comments and interesting questions.

Below are point-to-point answers.

  1. We have modified the text to use less active voice and more passive voice.

  2. As underlined in the Discussion, we don’t know yet whether the model will be of practical relevance to represent small-scale turbulence. We have designed it primarily as a toy model, that can be used to get insight on turbulence properties at small scale. We are actually planning to investigate in the future whether this model can be used to describe dynamics of extreme events of dissipation, but we do not yet have an answer.

Reviewer 3 Report

Please, see the atached review.
